# Reconstruction of Subsurface Salinity Structure in the South China Sea Using Satellite Observations: A LightGBM-Based Deep Forest Method

**Lin Dong [1], Jifeng Qi [2,3,4,\*], Baoshu Yin [2,3,4], Hai Zhi [5], Delei Li [2,3,4], Shuguo Yang [1], Wenwu Wang [6], Hong Cai [1] and Bowen Xie [1]**

[1]  School of Mathematics and Physics, Qingdao University of Science and Technology, Qingdao 266061, China; 2020090004@mails.qust.edu.cn (L.D.); ysg_qust@qust.edu.cn (S.Y.); caihong@qust.edu.cn (H.C.); 2021090032@mails.qust.edu.cn (B.X.)
[2]  CAS Key Laboratory of Ocean Circulation and Waves, Institute of Oceanology, Chinese Academy of Sciences, Qingdao 266071, China; bsyin@qdio.ac.cn (B.Y.); deleili@qdio.ac.cn (D.L.)
[3]  Pilot National Laboratory for Marine Science and Technology (Qingdao), Qingdao 266237, China
[4]  University of Chinese Academy of Sciences, Beijing 100049, China
[5]  College of Atmospheric Sciences, Nanjing University of Information Science and Technology, Nanjing 210044, China; zhihai@nuist.edu.cn
[6]  Department of Electrical and Electronic Engineering, University of Surrey, Guildford GU2 7XH, UK; w.wang@surrey.ac.uk
\*  Correspondence: jfqi@qdio.ac.cn

**Abstract:** Accurately estimating the ocean's interior structures using sea surface data is of vital importance for understanding the complexities of dynamic ocean processes. In this study, we proposed an advanced machine-learning method, the Light Gradient Boosting Machine (LightGBM)-based Deep Forest (LGB-DF) method, to estimate the ocean subsurface salinity structure (OSSS) in the South China Sea (SCS) by using sea surface data from multiple satellite observations. We selected sea surface salinity (SSS), sea surface temperature (SST), sea surface height (SSH), sea surface wind (SSW, decomposed into eastward wind speed (USSW) and northward wind speed (VSSW) components), and the geographical information (including longitude and latitude) as input data to estimate OSSS in the SCS. Argo data were used to train and validate the LGB-DF model. The model performance was evaluated using root mean square error (RMSE), normalized root mean square error (NRMSE), and determination coefficient ($R^2$). The results showed that the LGB-DF model had a good performance and outperformed the traditional LightGBM model in the estimation of OSSS. The proposed LGB-DF model using sea surface data by SSS/SST/SSH and SSS/SST/SSH/SSW performed less satisfactorily than when considering the contribution of the wind speed and geographical information, indicating that these are important parameters for accurately estimating OSSS. The performance of the LGB-DF model was found to vary with season and water depth. Better estimation accuracy was obtained in winter and autumn, which was due to weaker stratification. This method provided important technical support for estimating the OSSS from satellite-derived sea surface data, which offers a novel insight into oceanic observations.

**Keywords:** machine learning; ocean subsurface salinity structure; South China Sea; satellite remote sensing data

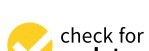



## 1. Introduction

Ocean salinity, a vital parameter of seawater, plays a significant role in understanding marine ecosystems, ocean dynamics, and climate changes [1–5]. For example, ocean salinity can be used as an indicator for the hydrologic cycle, which provides valuable insights into the understanding of global water cycle features [6–8]. Changes in ocean salinity may also play a role in the formation of water masses [9–11]. To better understand the role of

ocean salinity in dynamic ocean processes and climate changes, it is necessary to clarify the vertical structure of ocean salinity.

As the largest marginal sea of the Western Pacific, the South China Sea (SCS) has several straits along its border that connect to the Sulu Sea, the Java Sea, and the Indian Ocean (Figure 1a). The deepest water (around 5000 m) is found in the Eastern part of the SCS, while extended continental shelves (less than 200 m) have been found in the Western and Southern regions [12]. In the climatological mean, the sea surface salinity (SSS) in the SCS is north-south oriented: the SSS decreases from 34.0 psu in the north to 32.7 psu in the south (Figure 1b). The maximum SSS is in the Northern part of the SCS, which is related to the intrusion of the Kuroshio water through the Luzon Strait from the Pacific [13–15]. A low salinity tongue extends from the Southern part of the SCS, reaching as far as 10°N, which is closely related to the freshwater discharge from the Mekong and Rajang Rivers. Due to its special geographical location, the spatial distribution of the salinity in the SCS has significant features which are closely related to El Niño–Southern Oscillation (ENSO) [16–19], Asian monsoons, and the Pacific Western boundary current system [15,20]. Previous studies have suggested that the variability of the salinity in the SCS has a significant influence on the regional circulation and climate changes [21–23]. However, due to the lack of observations, little is known about the spatial and temporal variability of the salinity in the SCS. This has greatly limited the research on the thermohaline structures in the SCS. Therefore, it is of great importance to accurately retrieve the ocean subsurface salinity structure (OSSS), which remains a challenging problem for researchers.

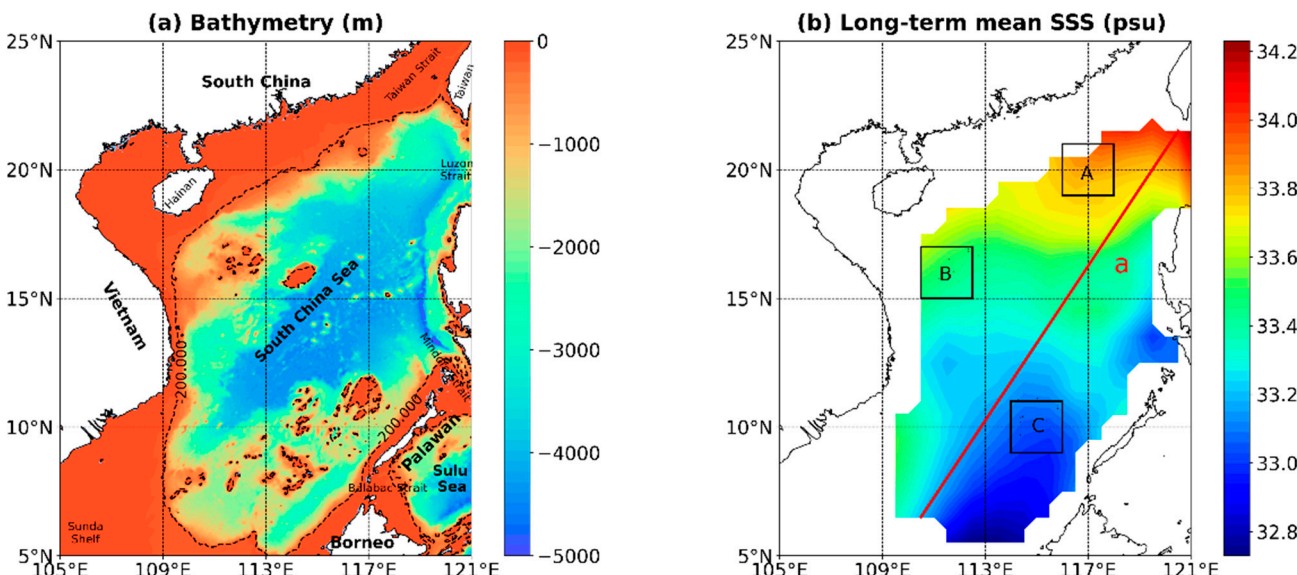

**Figure 1.** (**a**) Bathymetry (m) and geography of the SCS and (**b**) spatial distribution of long-term mean SSS (from January 2010 to December 2019) from Argo in the SCS. The three black boxes denote the study regions used in this study. The red line represents the location of transect used in this study.

Early studies on the estimation of ocean thermohaline structures in the SCS were usually based on numerical modeling and data assimilation [24–28]. For example, Chao et al. [24] modeled the interannual variations of thermal structure in the SCS by a three-dimensional primitive equation and found warming of the upper ocean during El Niño in the 1980s. Chu et al. [25] used the Princeton Ocean Model (POM) to investigate the seasonal variation of the thermal structure in the SCS. As researchers realized that the decreasing of dissolved oxygen was likely associated with the slowdown of thermohaline circulation, Li and Qu [26] analyzed the thermohaline circulation in the SCS on the basis of the available historical oxygen data. In order to provide better initial and boundary conditions for numerical simulations, assimilation methods have been used. Xiao et al. [27] performed an assimila-

tion experiment for the Southern SCS, and the altimeter data were assimilated into POM. Shu et al. [28] focused on correcting temperature in the mixed layer by projecting sea surface temperature (SST) onto subsurface observations based on the optimal interpolation in the SCS using the POM. Although numerical ocean models offer important tools for estimating ocean thermohaline structures, these dynamical models are computationally expensive, as simulating physical governing equations demands intensive computational resources.

In recent decades, remote sensing technology has experienced a remarkable and rapid advancement that has provided large amounts of useful satellite-derived sea surface data, such as SSS, SST, and sea surface height (SSH). These well-sampled surface observations have significantly improved our understanding of upper ocean dynamic processes. Although satellite observations have been confined to the surface, they can be used to infer information about the vertical structures of the ocean, such as temperature and salinity structures [29–34]. Previous studies suggested that many oceanic subsurface phenomena have surface manifestations [35–40]. For example, the SSH was determined by the seawater density field, and the overall integrated effect of thermohaline is constrained by the SSH, according to observations [36]. There was a high correlation between temperature and salinity variables in the ocean; that is, the vertical distribution of the salinity could be deduced from the SST [37]. The thermocline was associated with the warming or cooling of surface ocean water through seasonal warming and the surface stratification or upwelling in deeper waters caused by offshore seawater transport [38]. Vernieres et al. [39] and Lu et al. [40] have demonstrated that there is a close link between the SSS and subsurface salinity structures. A number of methods, such as linear regression of variables, and statistical and dynamic methods, have been used to estimate vertical ocean temperature and salinity structures using satellite-derived sea surface data [41–44]. For example, Carnes et al. [41] inferred the global subsurface thermohaline structure using SSH and SST through a least-squares regression method. Based on the empirical orthogonal function (EOF) method, Maes and Behringer [45] estimated the vertical salinity structure in the Western Pacific Ocean by using sea level anomaly (SLA) and SST. Chu et al. [46] proposed a parametric model based on a layered structure that successfully reproduced the subsurface thermal structure in the SCS using SST. A coupled pattern reconstruction (CPR) method was proposed for estimating the subsurface temperature profiles from SSH and SST, which was shown to provide a substantial improvement [47]. Guinehut et al. [44] successfully reconstructed global temperature and salinity fields at a high resolution based on sea surface data and in situ measurements through a linear regression method. Yang et al. [48] developed a new method based on a transfer function and a neural network to estimate vertical profiles of the salinity in the global ocean from the SSS observed by the Soil Moisture and Ocean Salinity (SMOS) satellite, which was reasonable in contrast with climatology. Considering the spatial non-stationarity feature, a satellite-based geographically weighted regression model was proposed to estimate the subsurface temperature anomaly (STA) of the Indian Ocean by combining satellite-derived sea surface data and Argo in situ data, which has a significant improvement over the linear regression model [34]. Although the estimation accuracy of subsurface thermohaline structures based on satellite-derived sea surface data was much better than that of the numerical model-based data assimilation, further improvements are possible.

With the rapid development of machine-learning technology, it has been extensively employed in the fields of ocean and atmosphere [49–52]. A number of machine-learning approaches, such as the artificial neural networks (ANN) [53–55], self-organization mapping (SOM) [56,57], support vector machine (SVM) [58,59], random forests (RF) [34,60,61], and extreme gradient boosting (XGBoost) [62], have been widely used to retrieve vertical thermohaline structures of the ocean. Ali et al. [53] used an ANN method to estimate the vertical thermal structure from SST, SSH, wind stress, net radiation, and net heat flux data. This model could successfully reconstruct the ocean subsurface thermal structure. The SOM neural network has been applied to SST, SSH, and SSS data to estimate the STA [56]. Considering the data space correlation, Chen et al. [57] combined the SOM method with

an EOF analysis to reconstruct the subsurface thermal structure by using the SST, the SSH, the longitude (LON), the latitude (LAT), and the month in the North-Western Pacific Ocean. Furthermore, machine-learning algorithms such as SVM, RF, and XGBoost were used to estimate the STA from surface remote sensing observations, which proved that the SSS and sea surface wind (SSW) were helpful in improving the accuracy of the estimations [34,58,60,62]. K-means clustering and feed-forward neural network were combined to estimate the subsurface temperature and achieve promising results in the deep ocean by taking the distribution of the ocean fields into consideration [63]. Based on a stacked long short-term memory (LSTM) neural network method, Buongiorno Nardelli [64] developed a model to estimate the ocean hydrographic profiles in the North Atlantic Ocean using surface remote sensing observations. Recently, Jiang et al. [65] proposed a bidirectional long short-term memory (Bi-LSTM) framework to estimate and analyze the subsurface temperature and salinity in the global ocean. A back-propagation neural network (BPNN) method was used to estimate the thermal structure in the North Pacific Ocean from sea surface data, such as SSH, SST, SSS, SSW, and sea surface velocity (SSV) [66].

As compared to temperature, relatively few attempts have been made to estimate OSSS from satellite-derived sea surface data using machine-learning methods [62,67,68]. For example, Gueye et al. [67] proposed a neural network model-based SOM for reconstructing salinity profiles of the tropical Atlantic Ocean from satellite-derived sea surface data. Salinity profiles in the Pacific Ocean can be estimated from satellite-derived sea surface data using a generalized regression neural network with the fruit-fly-optimization algorithm (FOAGRNN) [68]. Su et al. [62] proposed XGBoost for retrieving subsurface thermohaline anomalies of the global ocean, including the STA and the subsurface salinity anomaly (SSA). These existing studies focused on large-scale ocean regions or the global ocean.

To the best of our knowledge, in the SCS, there are no related studies conducted to estimate the OSSS from satellite-derived sea surface data using machine-learning methods. In this study, we proposed a Light Gradient Boosting Machine (LightGBM)-based Deep Forest (LGB-DF) method to estimate OSSS in the SCS from satellite-derived sea surface data, including SSS, SST, SSH, SSW (decomposed into eastward wind speed (USSW)) and northward wind speed (VSSW) components), and the geographical information (LON and LAT). To evaluate the performance of the LGB-DF model, another popular machine-learning model, LightGBM, was also used to estimate the OSSS in the SCS.

The rest of the paper is organized as follows. The data and methods are presented in Section 2. The evaluation of the model performance in estimating OSSS in the SCS is presented in Section 3. Finally, the discussion and conclusions are provided in Section 4.

## 2. Data and Method

### 2.1. Data

As an important part of the Indian–Western Pacific Ocean warm pool, salinity changes in the SCS play an important role in regulating the regional and global climate system [69]. Therefore, we selected the SCS (105°E–121°E and 5°N–23°N) as our study area.

In this study, we used two sources of ocean observational data: the sea surface data from satellite observations, such as SSS, SST, SSH, and SSW, combined with geographical information (LON and LAT); and gridded Argo data. The SSS data were obtained from the SMOS with a spatial resolution of 0.25° latitude × 0.25° longitude [70]. The SST data were obtained from the National Oceanic and Atmospheric Administration (NOAA), which consisted of optimal interpolated data observed by the satellite radiometer with a spatial resolution of 1° latitude × 1° longitude [71]. The SSH data were obtained from the Archiving, Validation, and Interpretation of Satellite Oceanographic data (AVISO) project with a spatial resolution of 0.25° latitude × 0.25° longitude [72]. The SSW data were obtained from Cross-Calibrated Multi-Platform (CCMP) gridded data, which are combined with multi-source data using a variational analysis method (VAM) to produce high-resolution (0.25° latitude × 0.25° longitude) gridded analyses [73]. The subsurface salinity data were obtained from the new version of the Roemmich–Gilson Argo Climatology (RG-Argo) data

with a spatial resolution of 1° latitude × 1° longitude [74], which includes 58 vertical levels, but only 44 levels were used as training labels as well as to evaluate the model performance on the estimation of the OSSS.

Considering the differences in data resolution and time period between the input and output data available in the SCS, all data used in this study were processed into monthly averaged data and interpolated to a resolution of 0.5° latitude × 0.5° longitude with the same coverage of the SCS and the time period from January 2010 to December 2019. It should be noted that, in order to ensure uniformity, data points were deleted if any variable was null at the same point. All the data used in this paper are shown in Table 1.

**Table 1.** Summary of the data used in this study.

| Index | Input Variable | Data Source | Output Variable | Data Source | Time Range | Time/Spatial Resolution |
|---|---|---|---|---|---|---|
| Data | SSS<br>SST<br>SSH<br>SSW | SMOS<br>NOAA<br>AVISO<br>CCMP | Salinity<br>(2.5–1000 m) | Argo | 2010–2019 | Monthly<br>0.5° × 0.5° |

*2.2. Method*

2.2.1. The LGB-DF Model

Deep Forest (DF) is an advanced decision-tree ensemble algorithm based on random forest, proposed by Zhou and Feng in 2017 [75,76]. Recently, the DF model has been widely used in many fields to prove its robustness in classification and prediction tasks [77–82]. A DF model would have great potential if it could go deeper. The LightGBM method has been shown to have the ability to estimate the ocean's subsurface information [83–87]. This inspired us to propose an improved DF model based on LightGBM (LGB-DF) method to estimate the OSSS in the SCS using satellite-derived sea surface data. In this study, the estimators (random forest and completely-random tree forests) of the DF model were replaced with the LightGBM to increase the accuracy of the model. The model code was based on the code of the open-source DF. The LGB-DF model was implemented and tested for all cases using Python programming on an Intel(R) Core(TM) I9-9940X CPU.

The flowchart for the proposed LGB-DF model is shown in Figure 2. The LGB-DF model had an important procedure: cascade structure, which could enhance the representational learning ability. In this study, the cascade structure of the LGB-DF model was constructed using two LightGBM (Figure 2). The number of trees in each forest was set to 150, and the maximum depth of each tree was set to 6. Having almost no adjustable hyperparameters was also one of the advantages of the LGB-DF model. As shown in Figure 2, the LGB-DF model processed the variables, layer by layer, in the cascade structure. In detail, the variables were input into the first layer, and each subsequent layer of input was spliced from the output of the preceding layer and the initial variable until the last layer to estimate the OSSS. To reduce the risk of overfitting, the vector produced by each estimator was generated by k-fold cross validation. Subsequently, the information in the last layer would be averaged as the estimation result. As compared to exiting machine-learning algorithms, the LGB-DF algorithm has the following advantages: fast speed, high accuracy, strong robustness, and simple implementation.

2.2.2. Experimental Setup

The flowchart of applying the LGB-DF model to estimate the OSSS is shown in Figure 3. The model setup was divided into three steps. The first step was the building of the training datasets. The satellite-derived sea surface data, such as SSS, SST, SSH, SSW (USSW and VSSW), and the geographical information (LON and LAT) were selected as input data for the LGB-DF model. The Argo data were used as training and testing labels. The second step was to train the model. The training data (from January 2010 to December 2018) were input into the LGB-DF model to obtain the output. Here, we used the grid search method to determine the optimal parameter combination for the LGB-DF model. Finally, with the



optimal parameter combination of the LGB-DF model, we estimated the OSSS in the SCS using sea surface data from the testing set (from January 2019 to December 2019).

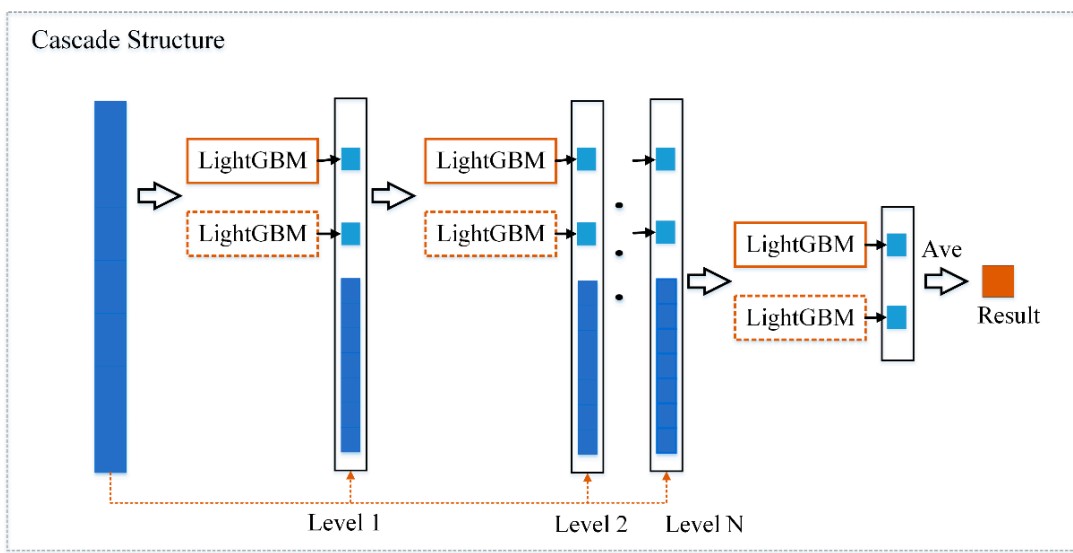

**Figure 2.** Flowchart for the LGB-DF model.

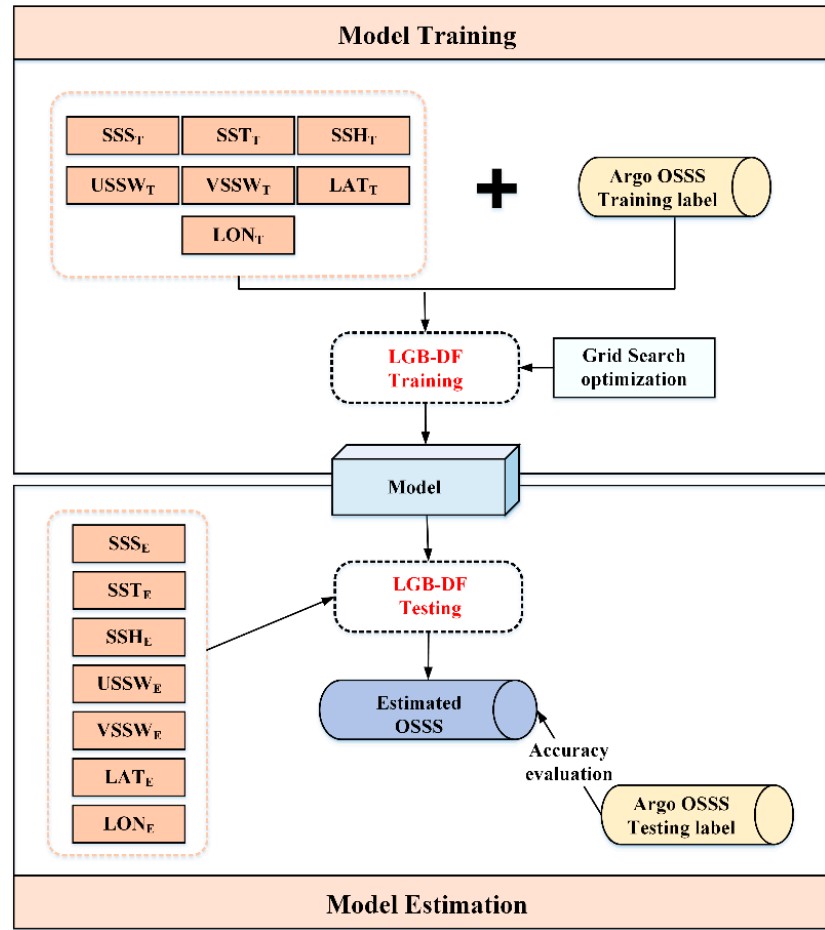

**Figure 3.** The flowchart of the OSSS estimation in the SCS using the LGB-DF model.

In addition to the LGB-DF model, we also set up a traditional machine-learning model (LightGBM) as a comparison to the LGB-DF model. Since the selection of input variables has an important impact on the performance of the model, three different combinations of sea surface parameters (three, five, and seven parameters) were used as LGB-DF model inputs to estimate the OSSS in the SCS. In this study, we evaluated the performance of the LGB-DF model through statistical metrics, such as root mean square error (RMSE), normalized root mean square error (NRMSE), and determination coefficient ($R^2$).

## 3. Results

### 3.1. Validation of Satellite-Derived SSS and SST

The accuracy of a machine-learning model is sensitive to the original input data [65]. Before utilizing the LGB-DF model to estimate OSSS in the SCS, the satellite-derived SSS and SST data were briefly validated by comparing them with the Argo data. As shown in Figure 4a, the seasonal variation of the satellite-derived SSS averaged over the SCS had good agreement with the Argo-derived SSS. For example, both of them showed that the maximum SSS value (>33.5 psu) occurred in April, and the minimum SSS value (<33.1 psu) occurred in November. The difference between the Argo SSS and satellite SSS varied from −0.02 psu to 0.14 psu. As for SST, the satellite-derived SST also showed good agreement with the Argo SST data on a seasonal scale (as shown in Figure 4b). In the SCS, the maximum SST value (>29.8 °C) occurred in May, whereas the minimum SST value (<26.3 °C) occurred in February. The difference between the Argo SST and satellite SST varied from −0.2 °C to 0.2 °C. Although the satellite data showed good agreement with the Argo observed data, some discrepancies were still observed, which may be due to different depths of measurement.

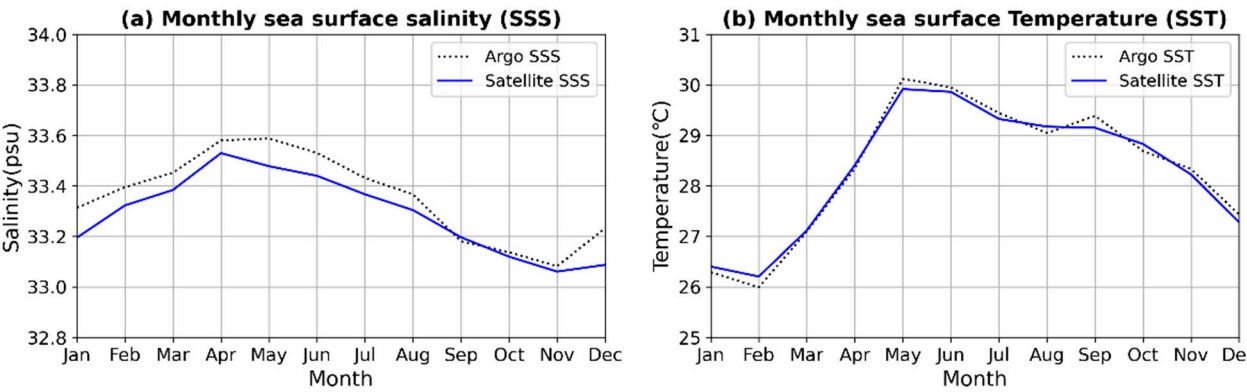

**Figure 4.** Comparison of the Argo (dashed black line) and satellite (solid blue line) for (**a**) the monthly mean SSS and (**b**) SST in the SCS from January 2010 to December 2019.

### 3.2. Identification of Input Variables

Previous studies had suggested that sea surface data could be used to infer ocean subsurface information with surface manifestations [37,55,62,86]. To determine the optimal combination of input variables for the LGB-DF model, a correlation analysis was conducted. Here, we only considered the absolute value of Pearson's correlation coefficients and focused on the magnitude of the correlation coefficients. The OSSS has a correlation with the sea surface variables at 50m, 100m, 500m, and 1000m of depth (Figure 5). The correlation coefficient between the OSSS and the SSS was relatively high at each depth, up to approximately 0.6. The correlation coefficients between the OSSS and SST/SSH/USSW (individually) were relatively small, approximately 0.2, while the VSSW was the lowest. As shown in Figure 5, the correlation coefficients between the OSSS and SSS/SSH/VSSW (individually) gradually decreased with depth, suggesting that SSS, SSH, and VSSW could play more important roles in the upper ocean. SST played a greater role in shallow and deeper layers, while the USSW performed better in the mid-ocean layers. The correlation analysis

between the OSSS and sea surface parameters at different depths elucidated the impact of SSS, SST, SSH, and SSW on OSSS and explained the reasons for the selected variables.

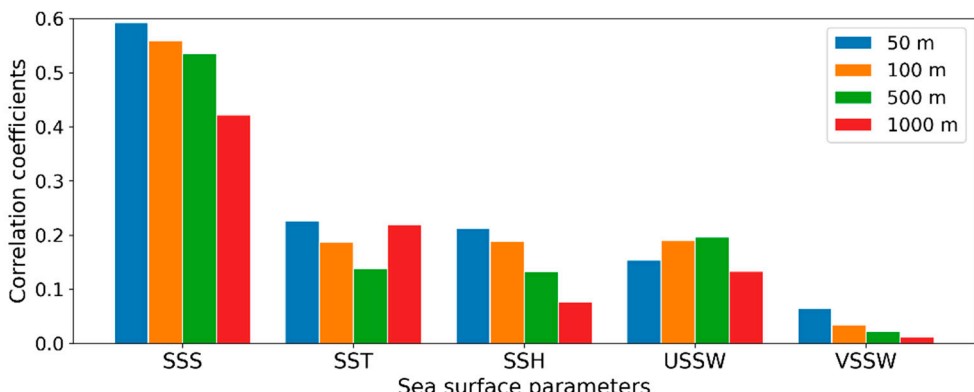

**Figure 5.** Correlation coefficients between the sea surface parameters (SSS, SST, SSH, USSW, and VSSW) and the Argo-observed OSSS at 50 m (blue), 100 m (orange), 500 m (green), and 1000 m (red) from January 2010 to December 2019.

As mentioned above, the unique geographical location and sparse observational data complicated the estimate of the OSSS in the SCS. Satellite-derived sea surface data captured most of the important features observed by the Argo surface data, providing an unprecedented opportunity to estimate OSSS in the SCS. Moreover, previous studies had suggested that geographical information could improve the estimation accuracy of the ocean subsurface information [67,86]. Therefore, we selected SSS, SST, SSH, SSW, and geographical information (LON and LAT) as the input variables to estimate the OSSS in the SCS.

### *3.3. Accuracy Comparison between the LGB-DF Model and LightGBM Model*

To illustrate the improved performance of the LGB-DF model, we compared the LGB-DF model to the LightGBM model in terms of RMSE and $R^2$. For the LGB-DF model and LightGBM model, the average RMSE and $R^2$ at all depth levels were 0.0320/0.9398 and 0.0398/0.9150, respectively. The OSSS estimated by the LGB-DF model had relatively lower RMSE and higher $R^2$ values not only on average but also at each depth level (Figure 6), indicating that the LGB-DF model was more accurate than the LightGBM model for the estimation of the OSSS in the SCS.

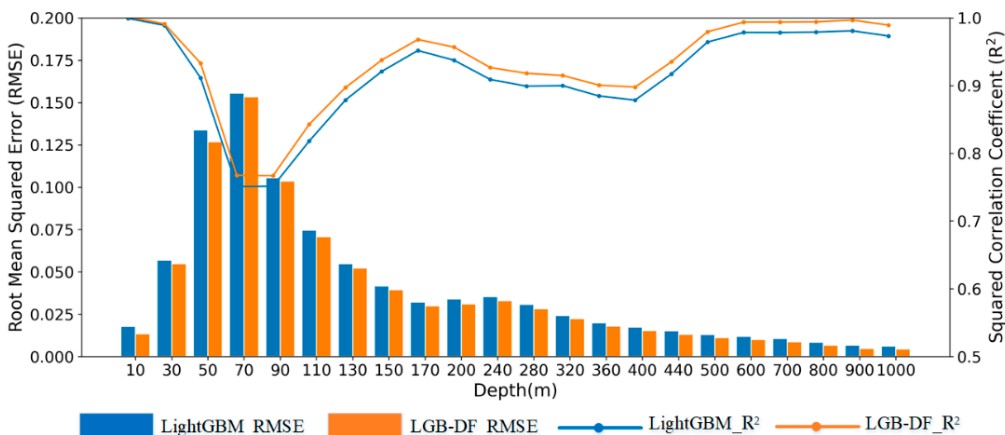

**Figure 6.** The average RMSE (psu) and $R^2$ of OSSS estimated using the LGB-DF model and LightGBM model at different depth levels in 2019 (the bars indicate the RMSE (psu) of the OSSS and the lines indicate the $R^2$ of the OSSS).

Another important issue in the estimation of the OSSS is the selection of input variables for models. Previous studies have suggested that the SSW and the geographical information could improve the accuracy of subsurface thermohaline estimates [62,67,85]. To further examine the influences of the SSW and the geographical information on the OSSS estimation in the SCS, we designed three sets of experiments with different input parameter combinations (Case 1, Case 2, and Case 3). In Case 1, we selected SSS, SST, and SSH as input parameters. In addition to the above parameters, we also selected SSW as an input parameter for Case 2. In Case 3, the geographical information (LON, LAT) was added as well as SSW.

The comparisons showed that both the SSW and the geographical information improved the estimation accuracy of the LGB-DF model in the SCS. The vertical mean RMSE and $R^2$ of the 7-parameter model in Case 3 were 0.0320 and 0.9398, respectively. For the 5-parameter model in Case 2, the vertical mean RMSE and $R^2$ were 0.0520 and 0.7569, respectively. For the 3-parameter model in Case 1, the vertical mean RMSE and $R^2$ were 0.0615 and 0.7150, respectively. The LGB-DF model in Case 3 (SSS, SST, SSH, USSW, VSSW, LON, and LAT) produced significantly lower RMSE values than the LGB-DF models in Case 1 (SSS, SST, and SSH) and Case 2 (SSS, SST, SSH, USSW, and VSSW) at all depths, while the $R^2$ values were higher than other cases (Figure 7). All these indicated that adding SSW and the geographical information significantly improved the estimation accuracy of the OSSS in the SCS using the LGB-DF model.

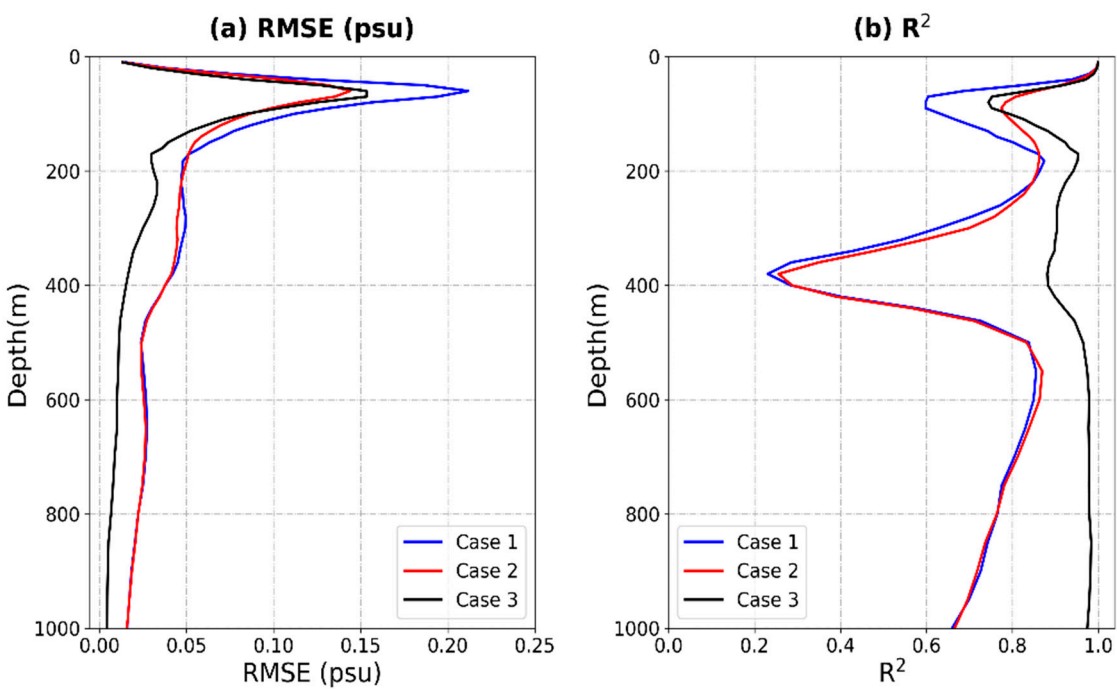

**Figure 7.** The estimation accuracy of the OSSS at different depths by the LGB-DF model based on (**a**) RMSE (psu) and (**b**) $R^2$ in different cases in 2019.

### 3.4. Evaluation of the LGB-DF Model

Based on the optimal parameter combination, the LGB-DF model was employed to estimate OSSS in the SCS. Next, we evaluated the performance and stability of the LGB-DF model from different aspects. Figure 8 shows the comparison of the LGB-DF-estimated OSSS and Argo-observed OSSS at depths of 50, 100, 500, and 1000 m in 2019; there were no significant differences between them. The LGB-DF model estimated OSSS showed good agreement with the Argo-observed OSSS at all depths. Most salinity features could be effectively reconstructed via sea surface data using the LGB-DF model. For example, at 50 m depth, both showed that there was a relatively high salinity tongue (>34.2 psu) in the northeast SCS. Relatively low salinity (<33.5 psu) was observed in the southeast

SCS (Figure 8a,e). The spatial distributions of the salinity at 100 m depth were similar to those at 50 m depth (Figure 8b,f). With increased depth, salinity tended to be stable. Below 500 m depth, the salinity varied from 34.4 psu to 34.6 psu. These spatial distribution features were well reconstructed by the LGB-DF model. From a horizontal point of view, the LGB-DF model had good performance in the estimation of OSSS in the SCS. More detailed descriptions of the LGB-DF model performance are discussed in the next section.

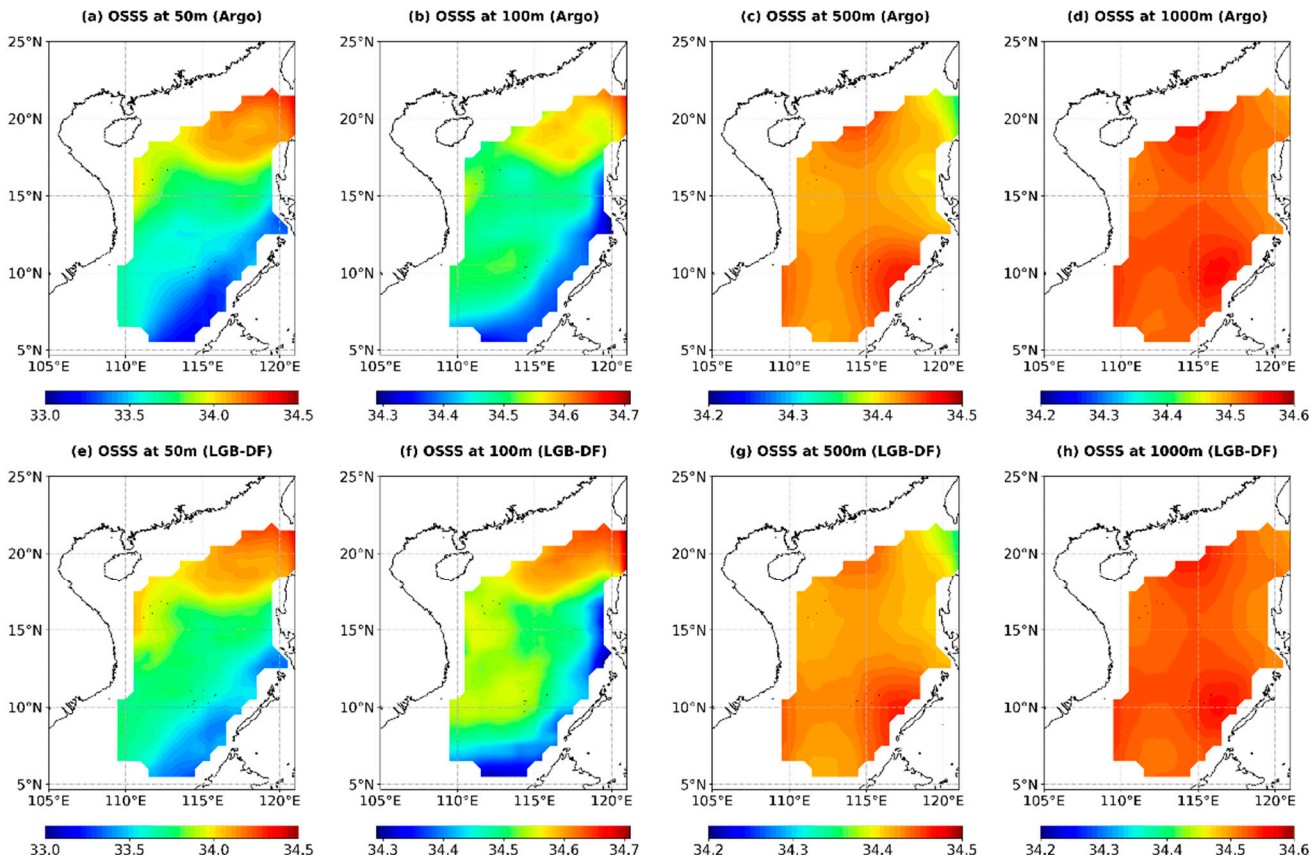

**Figure 8.** Argo-observed (**a**–**d**) and LGB-DF-estimated (**e**–**h**) yearly mean salinity at different depths (50, 100, 500, and 1000 m) in 2019.

To further evaluate the validity of the LGB-DF model, the accuracy of the OSSS estimation was quantitatively evaluated using the performance measures of RMSE and $R^2$ at different depths (Table 2). The RMSE could visually reflect the true errors at different depths. We employed the Argo-observed salinity at the same depth levels to validate the estimation results. As shown in Table 2, the RMSE value of the LGB-DF model exhibited differences at different depths; for example, RMSE = 0.1269 psu and $R^2$ = 0.9181 at 50 m depth, RMSE = 0.0841 psu and $R^2$ = 0.7919 at 100 m depth, RMSE = 0.0112 psu and $R^2$ = 0.9645 at 500 m depth, and RMSE = 0.0044 psu and $R^2$ = 0.9744 at 1000 m depth. The RMSE of the LGB-DF model decreased with depth due to the decreased range and standard deviation of the OSSS at deeper depths.

To improve the comparability of the model accuracy at different depths, we normalized the RMSE values to the relative error, i.e., NRMSE, dividing RMSE by the standard deviation of the Argo salinity at that depth. As shown in Figure 9, the NRMSE values increased from the surface to approximately 70 m, and then decreased from 70 m to 150 m, and then increased from 150 m to approximately 350 m, finally decreased from 350 m to 500 m, and stabilized from 500 m to 1000 m depth; whereas an opposite trend was observed in $R^2$. At approximately 70 m depth, the NRMSE value was the highest, while the $R^2$ was the lowest. This indicated that the estimation accuracy of the LGB-DF model was

lowest at approximately 70 m depth. This was likely due to 70 m being approximately the depth of the thermocline layer in the SCS [64], where the temperature and salinity changed more drastically with depth than in the layers above or below. This led to the difficulty of reconstructing the OSSS in the SCS. Although the estimation accuracy at approximately 70 m was relatively low, the LGB-DF model was generally satisfactory. This also suggested that the LGB-DF model could accurately estimate the OSSS of the SCS using satellite-derived sea surface data with satisfactory performance.

**Table 2.** Vertical distributions of RMSE (psu) and $R^2$ for the LGB-DF model at different depths in 2019.

| Depth (m) | RMSE | $R^2$ |
|---|---|---|
| 30 | 0.0547 | 0.9893 |
| 50 | 0.1269 | 0.9181 |
| 70 | 0.1533 | 0.7526 |
| 100 | 0.0841 | 0.7919 |
| 200 | 0.0310 | 0.9418 |
| 300 | 0.0249 | 0.9043 |
| 400 | 0.0153 | 0.8829 |
| 500 | 0.0112 | 0.9645 |
| 600 | 0.0100 | 0.9788 |
| 700 | 0.0087 | 0.9789 |
| 800 | 0.0066 | 0.9792 |
| 900 | 0.0047 | 0.9818 |
| 1000 | 0.0044 | 0.9744 |

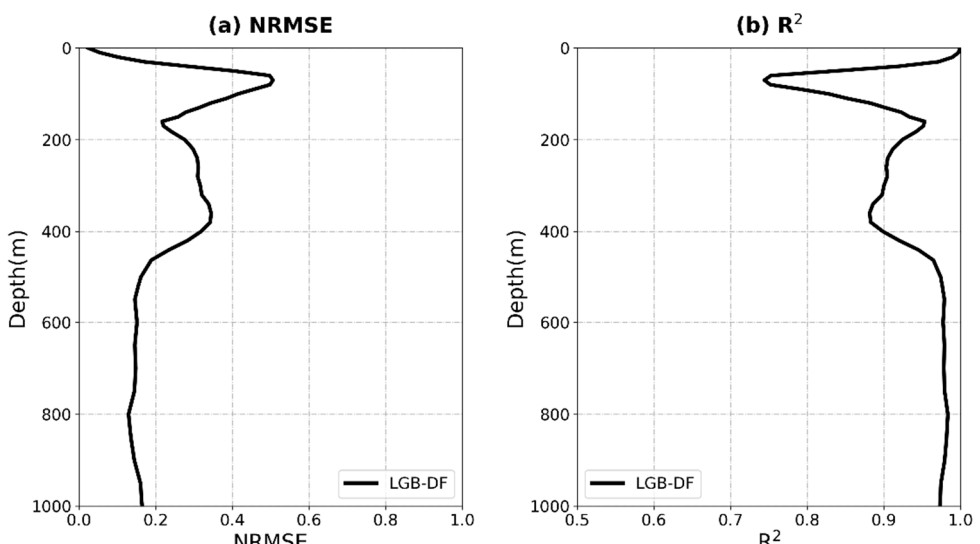

**Figure 9.** The estimation accuracy of OSSS in terms of NRMSE and $R^2$ at different depths by the LGB-DF model in 2019.

Next, to further evaluate the vertical performance of the LGB-DF model, we also compared the model estimated vertical salinity profiles with the Argo-observed salinity profiles in typical regions. Based on the characteristics of bathymetry and salinity distributions, we selected three typical boxes with a size of 2° × 2°, namely, Boxes A, B, and C (Figure 1). Box A (116°E~118°E and 19°N~21°N) was located along the continental slope south of China. Box B (110.5°E~112.5°E and 15°N~17°N) was situated in the region of the East Vietnam eddy. Box C (114°E~116°E and 9°N~11°N) was located in the Southern SCS. The vertical salinity profiles estimated by the LGB-DF model generally coincided with the Argo-observed profiles (Figure 10a–c). The vertically averaged RMSE and $R^2$ values between the LGB-DF estimation and the Argo observation were 0.0131 psu and 0.9950 for Box A,

0.0228 psu and 0.9942 for Box B, and 0.0594 psu and 0.9820 for Box C, respectively. Our comparison showed that the salinity difference between the LGB-DF estimation and the Argo observation for Box C was larger than those for Box A and Box B, and the maximum difference reached as high as 0.2 psu at approximately 70 m depth (Figure 10d). Although there were some differences, the LGB-DF model estimated salinity profiles were in good agreement with the Argo-observed salinity profiles. This result also demonstrated that the LGB-DF model was reliable and performed well in the estimation of OSSS in the SCS.

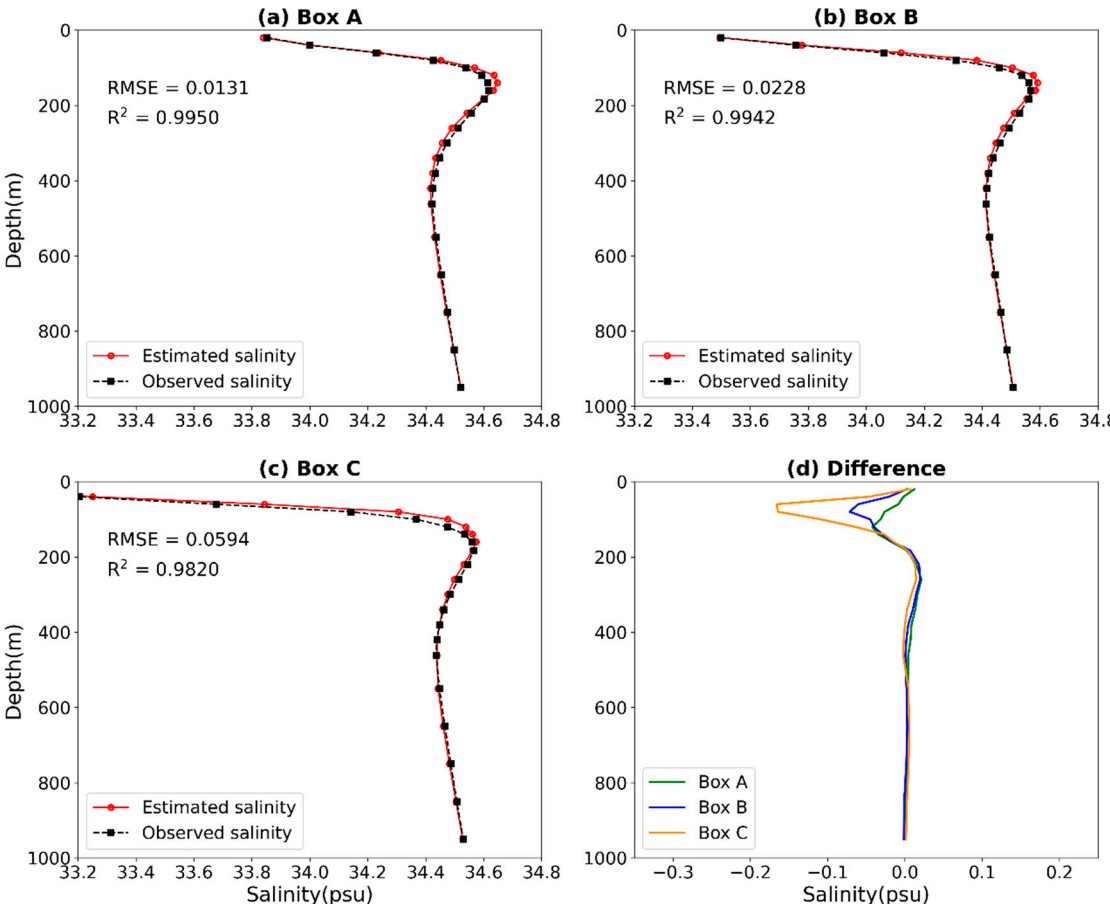

**Figure 10.** Comparison of the LGB-DF-estimated and Argo-observed salinity profiles averaged at different depths in Boxes (A–C) in 2019 (**a**–**c**) and their differences (**d**); Box A (116°E~118°E, 19°N~21°N), Box B (110.5°E~112.5°E, 15°N~17°N), and Box C (114°E~116°E, 9°N~11°N).

In addition, we selected a transect passing through the SCS from the southwest to the northeast to further evaluate the performance of the LGB-DF model. Figure 11 shows the comparison of the Argo-derived OSSS and LGB-DF model estimated OSSS in this transect. The results showed that the spatial distribution of OSSS from the LGB-DF model estimation was in good agreement with the Argo observations. Most of the observed significant features of the OSSS in this transect could be accurately reconstructed by the LGB-DF model. For example, in the upper 100 m, both of them showed that the salinity changed dramatically with depth, ranging from 33.1 psu at the surface to 34.5 psu at 100 m. The maximum salinity occurred between 100 m and 150 m in depth. Below 150 m, the salinity changed slightly but tended to be stable, ranging from 34.4 psu at 300 m depth to 34.6 psu at 1000 m depth. Figure 11c shows the salinity differences between Argo-observed and LGB-DF model estimated data (namely, Argo observation minus LGB-DF estimation). The results showed that the major differences (exceeding 0.25 psu) were present at a depth from 40 m to 150 m, between 9°N and 14°N, with Argo values less than the estimated salinity value; whereas Argo values more than the estimated salinity value were present at

a depth from 40 m to 150 m, between 16°N and 19°N. Overall, the spatial distribution of the salinity from the LGB-DF model estimation had a very similar pattern as compared to the Argo observations, further indicating that the LGB-DF model had good performance in the estimation of the OSSS in the SCS.

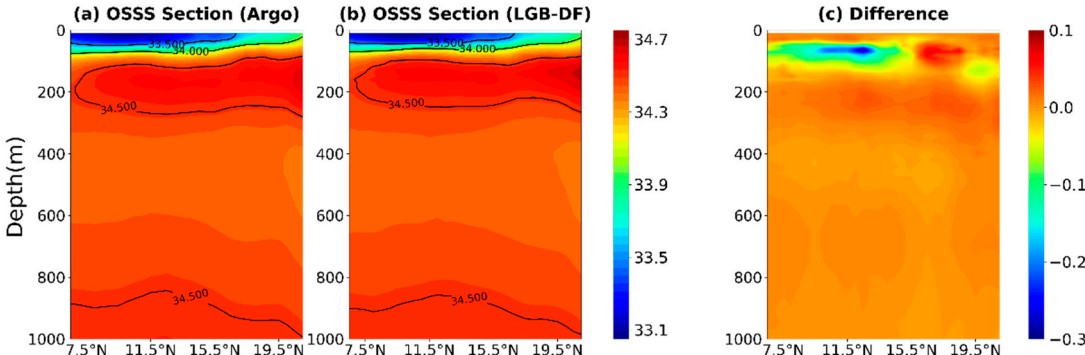

**Figure 11.** Argo-observed OSSS, LGB-DF-estimated OSSS and the differences between them, along a transect passing through the SCS from southwest to northeast. The black contours are, from top to bottom, 33.5 psu, 34.0 psu, and 34.5 psu, respectively.

The accuracy of the estimation by the model could also be evaluated directly using a density scatter plot. Therefore, we also calculated density scatter plots of the salinity from the Argo observations and LGB-DF model estimations to evaluate the performance of the LGB-DF model. The scatter distribution of the salinity from the LGB-DF estimation and the Argo observations at different depths for all geographical locations in 2019 is shown in Figure 12. Most of the scatter points were distributed evenly and densely along the line near 1:1 with a low RMSE. The RMSE values between the Argo-observed salinity and LGB-DF model estimated salinity were 0.0809 psu at 50 m depth, 0.0449 psu at 100 m depth, 0.0023 psu at 500 m depth, and 0.0012 psu at 1000 m depth. These also indicated that the estimated results by the LGB-DF model were reliable.

As previously discussed, the LGB-DF model had good performance in the yearly mean OSSS estimation in the SCS. However, the question of how it would perform in different seasons remained. In this study, we selected February, May, August, and November, all in 2019, to represent the winter, spring, summer, and autumn seasons of the year, respectively. Our quantitative evaluation of OSSS estimation for different seasons at the different depth levels (30, 50, 70, 100, 200, 300, 500, 600, 700, 800, 900, and 1000 m) in terms of the NRMSE and $R^2$ results are shown in Figure 13.

Generally, the NRMSE values in different seasons showed first an uptrend and then a downtrend, with a turning point appearing at 70 m. The highest NRMSE values occurred at 100 m in February and May at 0.3864 and 0.4085, respectively, and at 70 m for August (0.4603) and November (0.4587). The trend features of $R^2$ were unstable and fluctuated. They first fluctuated in the upper 500 m layer and then showed an uptrend from 500 m to 1000 m. The estimation accuracy of the LGB-DF model varied with the seasons. The average NRMSE ($R^2$) in February and November were 0.2052 and 0.2204 (0.9505 and 0.9377), respectively, which was lower (greater) than those in May and August. This indicated that the estimation accuracy in winter and autumn was better. The average NRMSE in May was 0.2676, and the average $R^2$ was 0.9112, which was the largest (smallest) value in four seasons. The average NRMSE and $R^2$ in August were 0.2646 and 0.9147, respectively. In general, the lower accuracy occurred in May and August, and the higher accuracy occurred in November and February, which could have been related to the different performances of the salinity at seasonal scales due to changes in the monsoonal circulation system. Specifically, the monsoon system dominated the summer pattern, and the winter pattern determined the climate of the SCS. The warm and humid southwest monsoon from the equator produced heavy precipitation and associated river runoff from mid-May to mid-

September, resulting in a double circulation pattern [14]. The dynamic ocean process was significant in May and August, resulting in poor model estimation. The results showed that there were low NRMSE and high $R^2$ values in all four seasons, indicating that the LGB-DF model had good seasonal applicability to estimate the OSSS in the SCS.

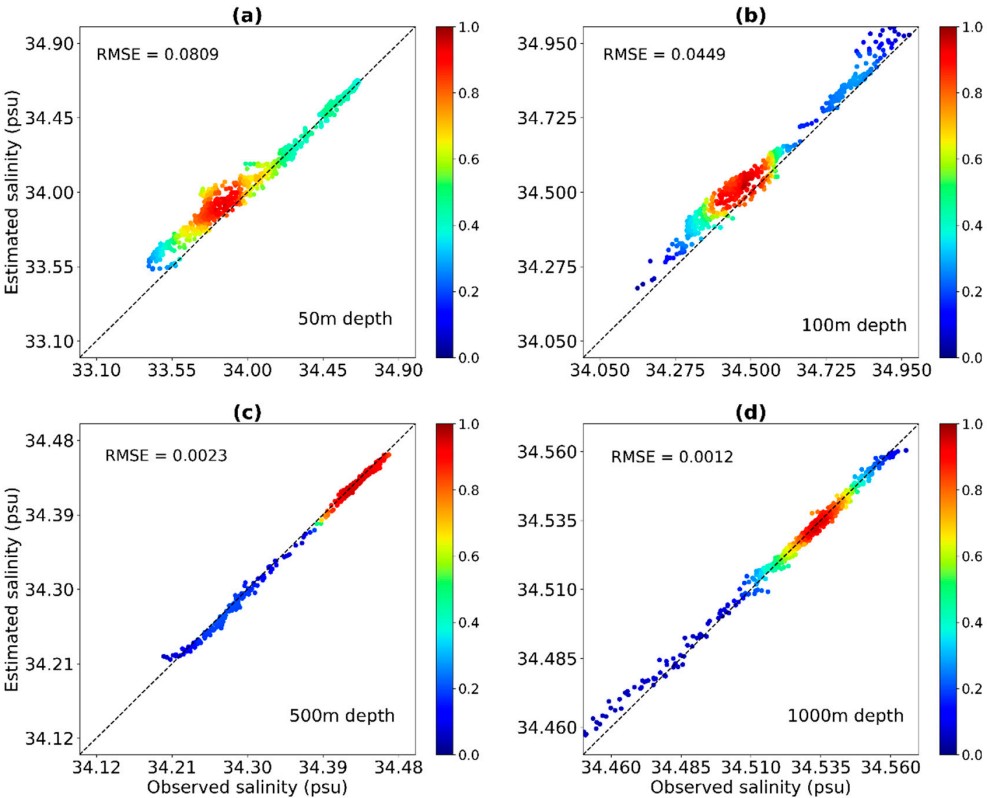

**Figure 12.** Density scatter plots of the salinity from LGB-DF estimations and Argo observations at (**a**) 50 m, (**b**) 100 m, (**c**) 500 m, and (**d**) 1000 m in 2019. The color bar represents the density of the scatter plots, with values closer to 1 indicating more scatter plots in the salinity range.

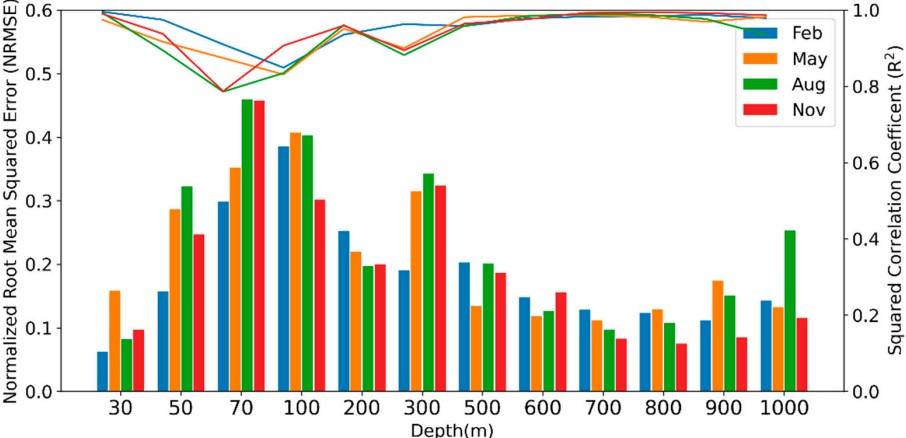

**Figure 13.** Performance measures by NRMSE and $R^2$ values of LGB-DF model for OSSS estimation at different depths in the SCS in 2019. Blue indicates February (winter), orange indicates May (spring), green indicates August (summer), red indicates November (autumn), the histograms display the NRMSE, and the lines display $R^2$.

## 4. Conclusions

Accurately estimating the vertical structure of ocean salinity in the SCS is of great importance for understanding oceanic processes due to its significant role in marine ecosystems, ocean dynamics, and climate changes. However, there is still a great lack of observational salinity data in the SCS due to the in situ observation being challenging and expensive. In this study, we proposed an LGB-DF model to estimate OSSS in the SCS. The developed LGB-DF model was used to reconstruct the OSSS in the SCS using satellite-derived sea surface data (SSS, SST, SSH, and SSW) and the geographical information (LON and LAT) as input data and in situ Argo data as label data. The LGB-DF-estimated results were measured for accuracy and reliability by RMSE, NRMSE, and $R^2$ using the Argo observational data.

Comparisons showed that the OSSS estimated by the LGB-DF model had relatively lower RMSE and higher $R^2$ values, not only on average but also at each depth level, as compared to the LightGBM model, indicating that the LGB-DF model accurately estimated the subsurface salinity of the SCS and outperformed the LightGBM model. This was attributed to the LGB-DF model combining the characteristics of deep learning and ensemble models to solve complex problems. In addition to SSH, SST, and SST, SSW and geographical information were two necessary parameters for accurately estimating the OSSS in the SCS and significantly improved the estimation accuracy of the LGB-DF model.

The results showed that the LGB-DF model had good performance in the estimation of the OSSS in the SCS with an area-averaged RMSE value of 0.0320 psu and an area-averaged $R^2$ value of 0.9398. The estimated salinity by the LGB-DF model and the Argo observed salinity both showed consistent spatial distribution at various depths in 2019. The performance measures showed that the performance of the LGB-DF model also varied with depth, with better performance in shallow layers due to the physical state relative to the surface being easily described. The performance of the LGB-DF model also varied with seasons: the average NRMSE ($R^2$) values in winter and autumn were lower (greater) than those in other seasons, indicating a better estimation accuracy was obtained in winter (NRMSE = 0.2052, $R^2$ = 0.9505) and autumn (NRMSE = 02204, $R^2$ = 0.9377). Although complex dynamic processes and the strong monsoon climate increase the difficulty of local OSSS estimation, our LGB-DF model had good performance in estimating the OSSS in the SCS according to satellite-derived sea surface data. This study demonstrated that the reconstruction of the subsurface salinity structure in the SCS using satellite observations based on the LGB-DF model was reliable and accurate.

Although the LGB-DF model has good applicability to estimate the vertical structure of the ocean salinity from the satellite-derived sea surface data, some discrepancies were observed in primarily two aspects. Data errors existed between the observed values and the true values due to objective factors such as the observation equipment itself and the environment. In the data processing, we interpolated the remote sensing data and Argo data to unify the resolution, which also caused errors. The estimation model error was also noted. The relationship between the sea surface data and the subsurface salinity could vary due to dynamic processes, such as subsidence and upwelling. Furthermore, as a data-driven method, the LGB-DF model was highly dependent on training data, which could underestimate or overlook the signal of some large anomalous events.

In future studies, we will further improve the estimation accuracy by using more accurate data and more advanced deep-learning methods combined with oceanic dynamic mechanisms to provide more explanatory results.

**Author Contributions:** Conceptualization, J.Q.; methodology, L.D.; validation, D.L., formal analysis, J.Q. and H.Z.; data curation, L.D.; writing—original draft preparation, L.D. and J.Q.; writing—review and editing, J.Q., S.Y. and B.Y.; visualization, B.X.; supervision, J.Q., W.W. and H.C.; All authors have read and agreed to the published version of the manuscript.

**Funding:** This research was funded by the Marine S&T Fund of Shandong Province for Pilot National Laboratory for Marine Science and Technology (Qingdao), grant number 2022QNLM010301-3,

National Natural Science Foundation of China, grant number 42176010, Natural Science Foundation of Shandong Province, China, grant number ZR2021MD022, Strategic Priority Research Program of the Chinese Academy of Sciences, grant number XDB42000000, National Natural Science Foundation of China, grant number 42076022.

**Data Availability Statement:** The datasets presented in this study are publicly available.

**Acknowledgments:** The SSS data were obtained from https://www.catds.fr/Products (accessed on 8 June 2021). The SST data were obtained from http://apdrc.soest.hawaii.edu/data/data.php (accessed on 6 June 2021). The SSH data were obtained from https://www.aviso.altimetry.fr/en/data/products (accessed on 12 June 2021). The SSW product was obtained from http://www.remss.com/measurements/ccmp/ (accessed on 16 June 2021). The gridded Argo data were obtained from http://sio-argo.ucsd.edu/RG_Climatology.html (accessed on 17 June 2021).

**Conflicts of Interest:** The authors declare no conflict of interest. The funders had no role in the design of the study, in the collection, analysis, or interpretation of the data, in the writing of the manuscript, or in the decision to publish the results.

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
