# Peer review of "Reconstruction of Subsurface Salinity Structure in the South China Sea Using Satellite Observations: A LightGBM-Based Deep Forest Method"

_remotesensing, doi:10.3390/rs14143494_

Round 1
Reviewer 1 Report
This manuscript proposed a classic and efficient LighGBM machine learning method for predicting subsurface salinity structure in the SCS. Although the method is effective with a good performance, this study really lacks methodological and experimental comparison, thus it is difficult to prove and determine the advantages of the proposed approach. This method has been proposed by the previous study for subsurface information estimation, and the novelty of the methodology should be enhanced. Moreover, the physical oceanography mechanisms for the projection of subsurface salinity from satellite remote sensing observations need further investigation and clarification. The extent to which satellite observations of sea surface parameters can provide useful information for estimating the subsurface salinity should be well figured out, why the author selected those surface parameters as the model input, and how important each surface parameter is for the subsurface salinity prediction?
I strongly recommend the author adopt more advanced AI-based techniques for comparison to make the methodology more robust, and meanwhile further validate the advantages and performance of the proposed method, and the oceanography mechanism or physical basis for the subsurface salinity prediction from sea surface variables in SCS should be well addressed. The machine learning model is a black box, but there should be some physical basis and explanation for the data-driven model construction for subsurface variable inversion based on satellite observations.
The performance measures for the model evaluation adopted here are only RMSE and R2, it is not enough for the performance validation and the NRMSE should be supplemented for a more reasonable accuracy assessment at different depth levels with varying OSSS value ranges.
I suggest the author update the technique framework and set up a contrastive experiment for model inter-comparison and validation. The uncertainties and limitations of the model should be further analyzed and discussed.
The input data only one year in 2019 are employed to reconstruct the OSSS, why not apply the model and extend the time series for more years for the data reconstruction, and analyzed and evaluated the time-series OSSS reconstruction dataset and its interannual evolution characteristics, as well as the data quality and accuracy assessment.
Author Response
Response to Reviewer #1’s Comments
Point 1: This manuscript proposed a classic and efficient LightGBM machine learning method for predicting subsurface salinity structure in the SCS. Although the method is effective with a good performance, this study really lacks methodological and experimental comparison, thus it is difficult to prove and determine the advantages of the proposed approach. This method has been proposed by the previous study for subsurface information estimation, and the novelty of the methodology should be enhanced.
Response 1: We sincerely appreciate this valuable comment, which has been very helpful in improving our manuscript. We fully agree with the reviewer that, enhancing the novelty of the methodology is very helpful to improve our manuscript. According to the reviewer’s comment, we have adopted a more advanced LightGBM-based Deep Forest (LGB-DF) method to estimate subsurface salinity structure in the SCS. For the LGB-DF model, the estimators (random forest and completely-random tree forests) of the DF model are replaced with LightGBM for increasing the accuracy of the model. In addition, we have added some discussion to prove the advantages of the proposed LGB-DF model by comparing with the LightGBM model. Please see lines 178-184, 228-256, 359-370 for more details.
Point 2: Moreover, the physical oceanography mechanisms for the projection of subsurface salinity from satellite remote sensing observations need further investigation and clarification.
Response 2: Thanks for this important suggestion, which is very helpful to improving the manuscript. We fully agree with the reviewer that it is necessary to further clarify the physical oceanography mechanisms for the projection of ocean subsurface salinity from satellite remote sensing observations. According to the reviewer’s comment, we have added some discussion in the revised manuscript to show that some sea surface observations can be used to infer information about the vertical structure of the ocean’s interior, like that of salinity and temperature structures. In addition, some new relevant references have been added in the revision (Stommel, 1947; Cornillon et al., 1987; Fiedler, 1988; Cooper and Haines, 1996; Vernieres et al., 2004; Lu et al., 2016). Please see lines 105-114 for more details.
References:
- Stommel, H. Note on the use of the TS correlation for dynamic height anomaly computations. Journal of Marine Research 1947, 6, 85-92.
- Cornillon, P.; Stramma, L.; Price, J.F. Satellite measurements of sea surface cooling during hurricane Gloria. Nature 1987, 326, 373-375. https://doi.org/10.1038/326373a0
- Fiedler, P.C. Surface manifestations of subsurface thermal structure in the California Current. Journal of Geophysical Research: Oceans 1988, 93, 4975-4983. https://doi.org/10.1029/JC093iC05p04975
- Cooper, M.; Haines, K. Altimetric assimilation with water property conservation. Journal of Geophysical Research: Oceans 1996, 101, 1059-1077. https://doi.org/10.1029/95JC02902
- Vernieres, G.; Kovach, R.; Keppenne, C.; Akella, S.; Brucker, L.; Dinnat, E. The impact of the assimilation of Aquarius sea surface salinity data in the GEOS ocean data assimilation system. Journal of Geophysical Research: Oceans 2014, 119, 6974-6987. https://doi.org/10.1002/2014JC010006
- Lu, Z.; Cheng, L.; Zhu, J.; Lin, R. The complementary role of SMOS sea surface salinity observations for estimating global ocean salinity state. Journal of Geophysical Research: Oceans 2016, 121, 3672-3691. https://doi.org/10.1002/2015JC011480
Point 3: The extent to which satellite observations of sea surface parameters can provide useful information for estimating the subsurface salinity should be well figured out, why the author selected those surface parameters as the model input, and how important each surface parameter is for the subsurface salinity prediction?
Response 3: We sincerely appreciate this insightful comment, which is very helpful to improving the manuscript. According to the reviewer’s comment, we have conducted a correlation analysis to clarify which satellite observations of sea surface parameters can provide useful information for estimating the subsurface salinity in the SCS, and how important each surface parameter is for the subsurface salinity prediction. A new section “3.2 Identification of input variables” has been added in the revised manuscript. In addition, a new Figure 5 has also been added in the revised manuscript. Please see lines 324-352 for more details.
Point 4: I strongly recommend the author adopt more advanced AI-based techniques for comparison to make the methodology more robust, and meanwhile further validate the advantages and performance of the proposed method, and the oceanography mechanism or physical basis for the subsurface salinity prediction from sea surface variables in SCS should be well addressed. The machine learning model is a black box, but there should be some physical basis and explanation for the data-driven model construction for subsurface variable inversion based on satellite observations
Response 4: Thanks for the important comment, which helped us to improve the quality of our manuscript. As suggested by the reviewer, we proposed an advanced AI-based technique (LGB-DF) to estimate the subsurface salinity structure in the SCS. To evaluate the performance of the proposed method, we have also made some comparisons with the LightGBM model. A new Figure 6 has been added in the revised manuscript. In addition, physical basis for the ocean subsurface salinity estimation from sea surface parameters has also been further investigated in the revised manuscript. There may be some deviations in subsurface salinity structure estimations at different depths and seasons. We have been given specific explanations in the revised manuscript. We have also made substantial changes in several part of the manuscript to address the reviewer’s concern. Hopefully you will find it justified. Please see lines 178-179, 228-256, 359-370, 556-561 for more details.
Point 5: The performance measures for the model evaluation adopted here are only RMSE and R2, it is not enough for the performance validation and the NRMSE should be supplemented for a more reasonable accuracy assessment at different depth levels with varying OSSS value ranges.
Response 5: Thanks for this important suggestion, which is also very important for this study. We fully agree with the reviewer that it would be better to use relative difference (i.e., normalized RMSE), instead of absolute difference, to evaluate the vertical performance of the LGB-DF model, which can improve the comparability of the model accuracy at different depths. According to the reviewer’s comment, we have used normalized RMSE (RMSE divided by the standard deviation of the Argo salinity at that depth) as measure to evaluate the vertical performance of the LGB-DF model. Figures 9 and 13 have also been revised. Hopefully you will find it justified. Please see lines 433-455, 565-569 for more details.
Point 6: I suggest the author update the technique framework and set up a contrastive experiment for model inter-comparison and validation. The uncertainties and limitations of the model should be further analyzed and discussed.
Response 6: Thanks for the important comment. As suggested by the reviewer, we have updated the technique framework and set up some contrastive experiments for model inter-comparison and validation. Two new Figures 2 and 3 have also been added in the revised manuscript to explain the process of the LGB-DF model.
In addition, the uncertainties and limitations of the model have been further analyzed and discussed in the revised manuscript. The uncertainties and limitations of the LGB-DF model mainly come from two aspects. On the one hand is the data error. There is a certain error between the observed value and the true value due to objective factors such as the observation equipment itself and the environment. In the process of data processing, we interpolate the remote sensing data and Argo data to unify the resolution, causing certain errors. On the other hand is the estimation model error. The relationship between the sea surface data and the subsurface salinity may vary due to different dynamic processes, such as subsidence and upwelling. Furthermore, as a data-driven method, the LGB-DF model is highly dependent on training data, which may underestimate and miss the signal of some large anomalous events. Please see lines 255-256, 275-300, 616-626.
Point 7: The input data only one year in 2019 are employed to reconstruct the OSSS, why not apply the model and extend the time series for more years for the data reconstruction, and analyzed and evaluated the time-series OSSS reconstruction dataset and its interannual evolution characteristics, as well as the data quality and accuracy assessment.
Response 7: Thanks for this question. It is well known that, successful utilization of a machine learning application requires data under the categories: training data, validation data, and testing data. The training data are used to train the model. The validation data are used to validate the performance of the model during training and control overfitting almost simultaneously. Due to the limitation of input data time period, in this study, we used 75 months of data for training, 33 months of data for validation and 12 months of data for testing. All the surface input data from January 2010 to December 2018 were randomly split into two separate categories: 70% of the dataset for training and the remaining 30% for validation. As input, the sea surface data from January 2019 to December 2019 were used to predict the OSSS in the SCS, of course, other years can also be used for testing.
Again, we sincerely appreciate the reviewer’s insightful comments and suggestions. We have worked hard in responding to these comments and suggestion. Thanks for your time and effort to help us improve the paper.
Reviewer 2 Report
My recommendation is "Reconsider after major revision”.
The manuscript is an interesting study about a new method for the reconstruction of subsurface salinity structure in the South China Sea using satellite observations.
In general, the manuscript is well organized and referenced, the applied methodology quite innovative and effective, and results important and relevant for the journal and the scientific community. However, in order to prove the greater accuracy of the proposed method, results from other similar studies using different algorithms should be discussed and compared to the ones of the present study. The discussion needs to be integrated accordingly. In addition, more information should be given about the software environment used for the proposed machine learning method (LightGBM) and statistical data analysis.
Further, the quality of English needs to be improved, so I recommend the manuscript to be sent to an English proofreading service before re-submission.
Finally, my review evidenced just a few suggestions for amendments, listed in the following section (“Specific comments”).
Specific comments.
Page 2, line 80: “tolls”? Misprint of “tools”?
Page 4, line 183: “monthly average data”. Why just monthly data? Any attempts with daily and/or at least weekly data?
Page 4, line 183: “…resolution of 0.5° latitude x 0.5° longitude”. Why, if the reported coarser resolution of input data is 0.25°?
Page 4, lines 185-186: “To improve the reliability of the LightGBM model and equally weight all input data, all 185 input data have been standardized as input data to the LightGBM model.” Try to reformulate, “input data” is repeated three time in the same sentence.
Page 5, line 195. Improve quality of Table 1. Contents inside are far to be clearly readable. Firstly, study area information could be reported in the text, and removed from the table. In addition, it is recommended to clearly separate input and “output” data (as far as concerns Argo data, I would say “training/test” data, or something like that, the actual output data are from the model estimation) data. Finally, a label for each column should be provided (i.e., for instance, “Input variable”, “Time range”, Time/spatial resolution”, “source of information”, etc.) .
Page 5, lines 201-214. Please provide details about the software used for the application of LightGBM algorithm. Is it R environment? If not, what else?
Page 5, line 218 and 221. Add “step” after “The first” and “The second”, respectively.
Page 8, line 283. Consider removing the term “clearly” after “LightGBM model”.
Page 8, line 285. Specify, at least in the caption, which data were used. Again from year 2019 similarly to Figure 4 and Figure 5?
Page 8, line 298. If I have correctly interpreted the meaning of the sentence, replace “its dose” with “it does”.
Page 9, line 326. Consider replacing “are generally coincide” with “are generally coincident”.
Page 10, line 335. Consider replacing ”are reliable” with “is reliable”, or amend the sentence accordingly (for instance, "the LightGBM model predictions are reliable…."
Page 10, line 344. Consider replacing “result shows” with “results show”.
Page 10, line 345. Consider replacing “shows a” with “is in”.
Page 10, line 353. Consider replacing “…estimated, which are computed by Argo observation minus LightGBM estimation.” with “…estimated (namely, Argo observation minus LightGBM estimation).”
Page 10, lines 357-358. Consider replacing “…the spatial distribution of salinity from LightGBM model estimation have very similar pattern with Argo observations” with “the spatial distribution of salinity from LightGBM model estimation has a very similar pattern compared to Argo observations”.
Page 11, line 361. Fix the misprint in the title of Figure 7(c) replacing "Diffreence" with "Difference".
Page 11, line 374 (Figure 8). Indicate the name of variable in the colorbars. What does they represent? Maybe R-squared? On which subsets of data are they calculated? In other words, what does each point represent? Single predictions in different greographical sites (by month?)?
Page 11, line 377. Check sentence “…as to how about…”.
Page 12, lines 386-388. Consider replacing with: "The highest RMSE value for the month of February occurs at 100 m (RMSE=0.0598), while for the other months at 50 m (May: 0.1232; August: 0.1436; November: 0.1112)."
Page 12, line 392. Consider replacing “less” with “lower”.
Page 12, line 395. “0.9602”. Is it a misprint? This value is equal to what reported for November at line 392.
Page 12, lines 396-397. Please check. Form the graph, at least average R-squared seems lower in May (orange line) rather than in August (green line).
Page 12, line 397. Consider replacing “The lower accuracy…” with “In general, the lower accuracy…”.
Page 12, lines 399-400. “…due to changes in the monsoonal circulation system.”. Too vague. Specify.
Page 12, line 400. “The results show that the model accuracy is relatively high…”. Relatively to what? Please provide some terms of comparison, also in support to what is stated in the discussion several times, which represents the main output of this paper.
Page 13, lines 440-442. “We hope that the results provided here represent a useful method to better reconstructing the vertical structure of ocean’s interior.” Reformulate. This should not be a “hope”. If the proposed method is better than others, results from other similar studies using different algorithms should be discussed and compared to the ones of the present study. Integrate the discussion accordingly.
Page 13, lines 442-444. “It should be noted, the LightGBM model, as a machine learning technique, has not yet been interpreted by the oceanic dynamic mechanisms.” What do the authors mean with that? Maybe that the applied algorithm does not integrate any ocean dynamic mechanisms? Please clarify.
Author Response
Response to Reviewer #2’s Comments
General Comments: The manuscript is an interesting study about a new method for the reconstruction of subsurface salinity structure in the South China Sea using satellite observations. In general, the manuscript is well organized and referenced, the applied methodology quite innovative and effective, and results important and relevant for the journal and the scientific community.
Response: We are grateful to the reviewer for the encouraging comments. We sincerely appreciate all valuable comments and suggestions, which have been very helpful in improving the manuscript. In the revision, we have taken all of them into account and tried our best to address each of them. We hope the revision is satisfactory to you.
Point 1: However, in order to prove the greater accuracy of the proposed method, results from other similar studies using different algorithms should be discussed and compared to the ones of the present study. The discussion needs to be integrated accordingly.
Response 1: Thanks for this important comment, which is very helpful to improving the manuscript. We fully agree with the reviewer that it is very important to add some comparisons with other different algorithms to prove the better accuracy of the proposed method. However, most existing studies related to estimation of OSSS from sea surface data have focused on large-scale areas, such as global and Indian Ocean, but no related studies have been carried out in the SCS. To address the concern of the reviewers, we have added some comparative experiments in the revised manuscript. We proposed an advanced AI-based technique (LGB-DF) to estimate the subsurface salinity structure in the revised manuscript. To evaluate the performance of the proposed method, we have also made some comparisons with the original LightGBM model. In addition, we set up three sets of experiments with different input parameter combinations (Case 1, Case 2, and Case 3) to examine the influence of different input parameters on the model. Two new Figures 6 and 7 have been added in the revised manuscript. We have also made substantial changes in several part of the manuscript to address the reviewer’s concern. Hopefully you will find it justified. Please see lines 228-256, 359-393 for more details.
Point 2: In addition, more information should be given about the software environment used for the proposed machine learning method (LightGBM) and statistical data analysis.
Response 2: We thank the reviewer for pointing this out. According to the reviewer’s comment, we have added suggested contents in the revised manuscript to give a brief description of the software environment used for the proposed machine learning methods and statistical data analysis. Please see lines 237-240.
Point 3: Further, the quality of English needs to be improved, so I recommend the manuscript to be sent to an English proofreading service before re-submission
Response 3: Thanks very much for pointing this out. Following this comment, we have gone through the revised manuscript and corrected some grammatical errors and typing mistakes. In addition, the paper has been edited by a professional language editing service provided by MDPI, we attached the language editing certificate along with manuscript.
Specific comments:
Point 1: Page 2, line 80: “tolls”? Misprint of “tools”?
Response 1: Revised as the reviewer suggested. Thanks.
Point 2: Page 4, line 183: “monthly average data”. Why just monthly data? Any attempts with daily and/or at least weekly data?
Response 2: Thanks for this question. In this study, we use the monthly gridded Argo dataset produced by Roemmich and Gilson (2009) as labeled training data, as well as validating the accuracy and reliability of the results from models. As a result, we only consider the monthly average data due to data limitations.
References:
- Roemmich, D.; Gilson, J. The 2004–2008 mean and annual cycle of temperature, salinity, and steric height in the global ocean from the Argo Program. Prog. Oceanogr. 2009, 82, 81–100.
Point 3: Page 4, line 183: “…resolution of 0.5° latitude x 0.5° longitude”. Why, if the reported coarser resolution of input data is 0.25°?
Response 3: Thanks for your question. In this study, the monthly gridded Argo data is used as labeled training data for machine learning models, which has a horizontal resolution of 1°×1° and is interpolated to 44 depth levels (upper 1000 m). Considering the differences in data resolution between the input (including SST, SSS, SSH, and SSW) and output data available in the SCS, all data used in this study have been processed into monthly mean and interpolated to a resolution of 0.5° latitude × 0.5° longitude with the same coverage of the SCS.
Point 4: Page 4, lines 185-186: “To improve the reliability of the LightGBM model and equally weight all input data, all input data have been standardized as input data to the LightGBM model.” Try to reformulate, “input data” is repeated three time in the same sentence.
Response 4: Thanks for the reviewer’s careful review. I'm very sorry for the confusion here. To address the reviewer’s concern, this sentence has been removed in the revised manuscript.
Point 5: Page 5, line 195. Improve quality of Table 1. Contents inside are far to be clearly readable. Firstly, study area information could be reported in the text, and removed from the table. In addition, it is recommended to clearly separate input and “output” data (as far as concerns Argo data, I would say “training/test” data, or something like that, the actual output data are from the model estimation) data. Finally, a label for each column should be provided (i.e., for instance, “Input variable”, “Time range”, Time/spatial resolution”, “source of information”, etc.) .
Response 5: Thanks for this important suggestion, which is very important for this study. As suggested by the reviewer, we have revised Table 1 in the revised manuscript. Please see lines 224-226.
Point 6: Page 5, lines 201-214. Please provide details about the software used for the application of LightGBM algorithm. Is it R environment? If not, what else?
Response 6: Thanks very much for pointing this out. Following this comment, we have added suggested contents in the revised manuscript to give a brief description of the software environment used for the proposed machine learning methods and statistical data analysis. Please see lines 237-240.
Point 7: Page 5, line 218 and 221. Add “step” after “The first” and “The second”, respectively.
Response 7: Revised as the reviewer suggested. Thanks for the reviewer’s careful review.
Point 8: Page 8, line 283. Consider removing the term “clearly” after “LightGBM model”.
Response 8: Revised as the reviewer suggested. Thanks.
Point 9: Page 8, line 285. Specify, at least in the caption, which data were used. Again from year 2019 similarly to Figure 4 and Figure 5?
Response 9: Revised as the reviewer suggested. Thanks.
Point 10: Page 8, line 298. If I have correctly interpreted the meaning of the sentence, replace “its dose” with “it does”.
Response 10: Revised as the reviewer suggested. Thanks.
Point 11: Page 9, line 326. Consider replacing “are generally coincide” with “are generally coincident”.
Response 11: Revised as the reviewer suggested. Thanks.
Point 12: Page 10, line 335. Consider replacing ”are reliable” with “is reliable”, or amend the sentence accordingly (for instance, "the LightGBM model predictions are reliable…."
Response 12: Revised as the reviewer suggested. Thanks.
Point 13: Page 10, line 344. Consider replacing “result shows” with “results show”.
Response 13: Thanks again for the reviewer’s careful review; it has been corrected in the revised manuscript.
Point 14: Page 10, line 345. Consider replacing “shows a” with “is in”.
Response 14: Thanks. It has been corrected in the revised manuscript.
Point 15: Page 10, line 353. Consider replacing “…estimated, which are computed by Argo observation minus LightGBM estimation.” with “…estimated (namely, Argo observation minus LightGBM estimation).”
Response 15: Thanks. This sentence has been rewritten as suggested by the reviewer.
Point 16: Page 10, lines 357-358. Consider replacing “…the spatial distribution of salinity from LightGBM model estimation have very similar pattern with Argo observations” with “the spatial distribution of salinity from LightGBM model estimation has a very similar pattern compared to Argo observations”.
Response 16: Revised as the reviewer suggested. Thanks.
Point 17: Page 11, line 361. Fix the misprint in the title of Figure 7(c) replacing "Diffreence" with "Difference".
Response 17: Thanks for the reviewer’s careful review; it has been corrected in the revised manuscript.
Point 18: Page 11, line 374 (Figure 8). Indicate the name of variable in the colorbars. What does they represent? Maybe R-squared? On which subsets of data are they calculated? In other words, what does each point represent? Single predictions in different geographical sites (by month?)?
Response 18: Thanks for this question. We apologize that our original statements were not clear. In this paper, we use the color bar to represent the density of the scatter plot (the number of points per unit area divided by the maximum value), with values closer to 1 indicating more scatter plots in the salinity range. The calculated subset of data is the salinity from the LGB-DF model estimation and the Argo observations at different depths for all geographic locations averaged in 2019 (yearly average). Among them, each point represents the salinity value of different geographical locations from the LGB-DF model estimation and the Argo observations in the SCS in 2019. Scatter plots of salinity from the Argo observations and the LGB-DF model estimation show that most of the points are more or less evenly distributed around the line of equal value, giving a low RMSE (Figure 12). These results also suggest that the estimated results of the LGB-DF model are reliable and performed well in the OSSS estimation in the upper 1000 m.
To address the reviewer’s concern, these sentences have been rewritten in the revised manuscript. Please see lines 512-514, 525-526 for more details.
Point 19: Page 11, line 377. Check sentence “…as to how about…”.
Response 19: Thanks for the reviewer’s careful review; this sentence has been corrected in the revised manuscript.
Point 20: Page 12, lines 386-388. Consider replacing with: "The highest RMSE value for the month of February occurs at 100 m (RMSE=0.0598), while for the other months at 50 m (May: 0.1232; August: 0.1436; November: 0.1112)."
Response 20: Revised as the reviewer suggested. Thanks very much.
Point 21: Page 12, line 392. Consider replacing “less” with “lower”.
Response 21: Thanks very much. It has been corrected in the revised manuscript.
Point 22: Page 12, line 395. “0.9602”. Is it a misprint? This value is equal to what reported for November at line 392.
Response 22: Thanks for the reviewer’s careful review. We propose a new method to estimate the subsurface salinity structure in the SCS, the corresponding results have been corrected in the revised manuscript.
Point 23: Page 12, lines 396-397. Please check. Form the graph, at least average R-squared seems lower in May (orange line) rather than in August (green line).
Response 23: Thanks for the reviewer’s careful review; I'm very sorry for the confusion here. There is a misprint for the May results. We corrected it in the revised manuscript.
Point 24: Page 12, line 397. Consider replacing “The lower accuracy…” with “In general, the lower accuracy…”.
Response 24: Thanks. Revised as the reviewer suggested.
Point 25: Page 12, lines 399-400. “…due to changes in the monsoonal circulation system.”. Too vague. Specify.
Response 25: Thanks for pointing this out, which is very important for the present study. As suggested by the reviewer, we have rewritten these sentences in the revised manuscript. Please see lines 556-561 for more details.
Point 26: Page 12, line 400. “The results show that the model accuracy is relatively high…”. Relatively to what? Please provide some terms of comparison, also in support to what is stated in the discussion several times, which represents the main output of this paper.
Response 26: Thanks for pointing this out. We apologize that our original statements were not clear. In the revision, we have rewritten this sentence according to the reviewer’s comment. Please see liens 561-564.
Point 27: Page 13, lines 440-442. “We hope that the results provided here represent a useful method to better reconstructing the vertical structure of ocean’s interior.” Reformulate. This should not be a “hope”. If the proposed method is better than others, results from other similar studies using different algorithms should be discussed and compared to the ones of the present study. Integrate the discussion accordingly.
Response 27: Thanks for the important comment. As suggested by the reviewer, we have added the following contents in the revised manuscript. Please see lines 584-589.
“Comparisons showed that the OSSS estimated by the LGB-DF model had relatively lower RMSE and higher R2 values, not only on average but also at each depth level, as compared to the LightGBM model, indicating that the LGB-DF model accurately estimated the subsurface salinity of the SCS and outperformed the LightGBM model. This was attributed to the LGB-DF model combining the characteristics of deep learning and ensemble models to solve complex problems.”
Point 28: Page 13, lines 442-444. “It should be noted, the LightGBM model, as a machine learning technique, has not yet been interpreted by the oceanic dynamic mechanisms.” What do the authors mean with that? Maybe that the applied algorithm does not integrate any ocean dynamic mechanisms? Please clarify.
Response 28: We apologize for the incorrect statement and thank the reviewer for pointing this out. We agree with the reviewer that there was some misleading information in the original manuscript. Therefore, it has been deleted in the revised manuscript to address the reviewer’s concern.
Again, we really appreciate the reviewer’s insightful comments and suggestions. We have worked hard in responding to these comments and suggestion. We deeply appreciate your help on improving the readability of our paper.

Round 2
Reviewer 1 Report
The manuscript has been well modified according to my comments and suggestions, and it can be accepted for publication now.
Author Response
Response to Reviewer #1’s Comments
Point 1: The manuscript has been well modified according to my comments and suggestions, and it can be accepted for publication now.
Response 1: We sincerely appreciate the reviewer’s insightful comments and suggestions. Thanks again for your time and effort to help us improve the paper.
Reviewer 2 Report
After reading the revised ms I think that, in general, authors have addressed most of the comments or issues that I mentioned in my review, including the need for improving the editing of English language.
A few additional suggestions for text editing are reported below.
Lines 113, 120, 486, 490: Use Italics for "in situ" .
Line 508: Consider replacing "the average NRMSE (R2) in winter and autumn were less (greater) than..." with "the average NRMSE (R2) values in winter and autumn were lower (greater) than...".
Author Response
Response to Reviewer #2’s Comments
General Comments: After reading the revised ms I think that, in general, authors have addressed most of the comments or issues that I mentioned in my review, including the need for improving the editing of English language.
Response: We are grateful to the reviewer for the encouraging comments. Again, we sincerely appreciate the reviewer’s insightful comments and suggestions, which have been very helpful in improving our manuscript.
Point 1: Lines 113, 120, 486, 490: Use Italics for "in situ" .
Response 1: Thanks. Revised as the reviewer suggested. Please see lines 113, 120, 486, 490.
Point 2: Line 508: Consider replacing "the average NRMSE (R2) in winter and autumn were less (greater) than..." with "the average NRMSE (R2) values in winter and autumn were lower (greater) than...".
Response 2: Thanks very much for pointing this out. This sentence has been revised according to the reviewer’s comment. Please see lines 508.
Again, we really appreciate the reviewer’s insightful comments and suggestions. We deeply appreciate your help on improving the readability of our paper.